# Helicopter Main Rotor Blade Parametric Design for a Preliminary Aerodynamic Analysis Supported by CFD or Panel Method

**DOI:** 10.3390/ma15124275

**Published:** 2022-06-16

**Authors:** Jakub Kocjan, Stanisław Kachel, Robert Rogólski

**Affiliations:** 1Doctoral School, Military University of Technology, ul. gen. Sylwestra Kaliskiego 2, 00-908 Warsaw, Poland; 2Faculty of Mechatronics, Armament and Aerospace, Military University of Technology, ul. gen. Sylwestra Kaliskiego 2, 00-908 Warsaw, Poland; stanislaw.kachel@wat.edu.pl (S.K.); robert.rogolski@wat.edu.pl (R.R.)

**Keywords:** helicopter main rotor, rotor blade, geometric modeling, aerodynamic panel method, CFD

## Abstract

This work is the preliminary part of a research program which is aimed at finding some new methods and design solutions for helicopter main rotor multidisciplinary optimization. The task was to develop a parametric geometric model of a single-blade main rotor applicable for varied methods of numerical aerodynamic modeling. The general analytical assumptions for the parametric main rotor design were described. The description of the main rotor blade parametric design method based on Open GRIP graphical programming was presented. Then, the parametric model of a blade was used for aerodynamic models independently developed for panel method and advanced CFD solver. The results obtained from the CFD simulations and panel analysis for main rotor aerodynamics were compared and assessed using analytical calculations. The calculations and simulations for a single-blade and completed rotor were performed for different helicopter weights and rotor pitch angles. The results of different computer aerodynamic analysis environments were compared for the possibility of their application in an optimization loop. This is preliminary work that describes only a partial problem that could be used in the future as part of a comprehensive methodology for aerodynamic and structural optimization of a helicopter rotor. As an output of the research, new options for main rotor optimization are developed. The combined parametric modeling with aerodynamic analysis, as described in this paper, provide the preliminary design for a main rotor spiral, as an element of the optimization loop.

## 1. Introduction

Military rotorcrafts that are currently operating are mostly constructions that were designed in the second half of the twentieth century. The constructions (in the preliminary and detail design phases) were prepared with classic mathematical calculations, eventually supported with CAD. With improved computerized options, design processes have been fully transferred to virtual environments. New aircraft structures are fully designed using computer methods; however, the structures of the existing constructions have also been modeled using engineering software, which is described in [1].

The most popular engineering design solution using CAD is a point and click method. A more comprehensive design method is to prepare a model that is generated as a result of entered parameters, which can be obtained by using scripting in the environment’s language; an example of parametric blade optimization is shown for a wind turbine blade in [2], and for a main rotor blade in [3]. The parametric design includes options to quickly change the model’s features without time-consuming 3D modeling. It is important to note that the influence of parametric change can be rapidly checked in numerous variants.

The helicopter rotor blade is an element which can be successfully adapted to parametric design, because of mathematical functions which precisely describe blade features. The parametric model is the first step to prepare the main rotor optimization process [4]. Some examples have been shown in different works: parametric airfoil design [5,6,7], structural design [8,9], and aerodynamic design [10]. A good parametric analysis of the rotor blade framework is presented in [11]. In this research, the parametric model was prepared using GRIP modeling language, which is an integral part of the Siemens NX environment. The usage of GRIP in parametric modeling was conducted in [12,13,14].

The proposed graphical programming language offers the possibility to prepare a geometric and structural model of the aircraft element. The designed part is fully described by the program script. In addition, the elements’ features can be calculated from mathematical functions, for example, polynomials for shape parametrization. After programming the structure, Open GRIP offers the possibility to analyze the inertia physiognomy of the body. The obtained data are the basis for the strength calculation. The program also grants a function for preparing forms which can be filled by the user. Therefore, some parameters can be changed during the design loop, without changing the program code. In the future, it is crucial to prepare a full parametric optimization. An effective approach to main rotor blade optimization was proposed in [8,10,15,16,17,18], and effective optimization of rotor construction was proposed in [19].

This work is part of a research program that is aimed at finding the best rotorcraft construction optimization solutions. The first step of the research program was an analysis of modern helicopter constructions and an outline of design parameters that are crucial in helicopter main rotor design. The results of the analysis showed that there is a field were rotor construction can be improved in accordance with modern combat field requirements which are dynamically changing due to fast technological improvements. The results, with charts and a comparison of parameters were published in [20]. 

On the basis of construction comparisons, we conducted a study of new design methods or combinations of some separate solutions. The aim of the study was to provide a versatile, accurate main rotor model that could be analyzed, validated, as well as quickly and easily changed in the optimization process. As a new approach to rotorcraft design, we combined the parametric programming CAD model generator with the CFD and panel method, with the goal of providing new possibilities for improving the construction optimization procedure. The effects and possibilities of CFD main rotor studies that were implemented in the main rotor development procedure were well presented in [21,22,23]. In addition, a CFD analysis, as a base for a strength analysis because of load prediction, was clearly shown in [24].

The development of the method started with the main rotor blade, as the main element that generates thrust. An existing rotor blade model was taken as a basis to conduct and authenticate the method. As previously described in the Introduction, the blade model was prepared using the parametric method in GRIP language. GRIP language was chosen because it is user-friendly graphical programming code for easily preparing the model with extensive built-in possibilities for quickly calculating the inertia parameters of the generated model, which could be used in further optimization steps, which is crucial when the strength of the construction is taken into consideration in the optimization loop. Further, the model was validated using the CFD and panel methods, and the thrust results were compared with the analytical calculations. The methods were also compared with each other to choose the appropriate procedure to perform the most efficient optimization. To the best of our knowledge, similar studies that combined the proposed methods for rotorcraft main rotor design and main rotor optimization procedure have not been conducted. 

This paper is organized as follows: The research methods are described in Section 4, where the mathematical model, which is the basis for the parametric programming, is defined, and in the second part of this section, the CFD and panel method settings are given; the validation results are described in Section 3; and the results are discussed and concluded in Section 4.

## 2. Research Methods

### 2.1. Parametric Description of a Single-Blade Rotor with Aspect of Its Geometrics and Aerodynamics

The construction process of helicopter rotor blades requires consideration of fatigue, strength, stiffness, cost, and vibrations. The rotor blade works in a changeable environment, and therefore, the designer has to take into consideration all of the parameters that transform during flight. Periodic changes in the flow velocity on the blade section can modify the angle of attack and the Mach number, and as a consequence, can change the aerodynamic coefficients. Blade turns (relative to hinge axis) and blade deformation cause a periodic change in inertial forces and provoke a cyclic modification in the angles of attack and flow velocity. The flexibility of the main rotor blades and their work in a strong centrifugal force field induce an inseparable connection between loads and deformations. An analysis of the problems mentioned and also the noise level or flow disturbance requires a versatile evaluation of the main rotor blade’s work conditions. Effective blade aerodynamic modeling was proposed in [25].

The precise blade geometric design is crucial for obtaining the required main rotor power. As described in [26], the geometric parameters of the blade are the following:-**Radius** The length of the blade measured from the axis of rotation to the tip;-**Chord** The length of the blade measured from the airfoil leading edge to the trailing edge and, for a tapered blade, it is the function of local radius;-**Airfoil** The cross-sectional shape that determines the aerodynamic parameters of the blade, i.e., lift, drag and momentum coefficients;-**Contour shape** This is the final shape of the blade, which depends on the chord function and tip shape;-**Geometric twist** Variation of the airfoil angle between the chord and a plane of rotation along the radius;-**Aerodynamic twist** Variation of the airfoil shape along the radius;-**Position and shape of trim tabs** Defines the possibility to adjust the rotating blade to the plane of rotation.

The area of a blade root airfoil transfers the whole load from the rotating aerodynamic surface to the blade grip sticking out of a rotor head block. The shape of the root is dictated by the build conception and the adopted aerodynamic solutions. The geometric twist is usually settled on several degrees. The shape of the rotor tip depends on the aerodynamic problems, for example, achieving the speed of sound or the noise level. An analysis of rotor blade tips was described in [27]. According to the mentioned work, there are three main types of helicopter tip designs: BERP tip, the parabolic tip, and the swept (tapered) tip. An example of a main rotor blade is shown in Figure 1.

To evaluate the prepared parametric model, a theoretical lift calculation must be conducted. The lift can be calculated in vertical flight using the blade element theory combined with the momentum theory. The first assumptions are the mass conservation equation for rotor disc:(1)$$\dot{m}=\rho Av_{i}$$
and the momentum conservation equation for hoovering rotor:(2)$$T=\dot{m}w$$

The energy conservation equation is the work of the rotor to change the rate of energy in fluid:(3)$$Tv_{i}=\frac{1}{2}\dot{m}w^{2}$$

The calculation is based on the thrust coefficient, as a function of pitch angle and inflow angle, and is given [28,29]:(4)$$C_{T}=\int_{0}^{1}\frac{a\sigma }{2}\left(\vartheta -\delta \right)r^{2}dr$$
where *a* is the lift curve slope, σ is the rotor solidity, ϑ is the pitch angle, δ is the inflow angle, and r is the current radius.

According to Figure 2, the inflow angle *δ* can be defined by:(5)$$\delta =arctg\left(\frac{W+v_{i}}{\Omega r}\right)$$
where W is the vertical velocity, $v_{i}\;$is the induced velocity, and Ω is the rotor angular velocity.

To calculate the induced velocity, for required thrust in hover, the equation solved from the momentum theory is given:(6)$$v_{i}=\sqrt{\frac{T}{2\rho A}}$$
where *T* is the rotor thrust, ρ is the air density, and *A* is the rotor planform area. For hover, the *W* velocity equals 0. 

The inflow angle can be defined with the inflow ratio as $\lambda_{i}=\delta /r$, therefore, the thrust coefficient will transform to:(7)$$C_{T}=\int_{0}^{1}\frac{a\sigma }{2}\left(\vartheta r^{2}-\lambda r\right)dr$$

Rotor performance can be calculated with the combination of the blade element theory with the momentum theory. Therefore, the induced velocity can be determined for nonhomogeneous inflow distribution, by using the differential form of the momentum theory:(8)$$dC_{T}=4\lambda_{i}\lambda rdr$$
with the blade element theory $C_{T}$ equation form. 

As a result, for hover where $\lambda_{c}=0$, the inflow ratio for the induced velocity is:(9)$$\lambda_{i}=\frac{a\sigma }{16}\left[\sqrt{1+\frac{32}{a\sigma }\vartheta r}-1\right]$$

The σ rotor solidity equation is necessary, which, in this work, is nonuniform along the blade span as a consequence of non constant chord. The rotor solidity is calculated as:(10)$$\sigma =\int_{0}^{1}\frac{bc\left(r\right)}{\pi R}dr$$
where *b* is the number of blades, *c*(*r*) is the chord in function of radius, and *R* is the rotor radius.

The thrust coefficient is the function of radial location of the blade measured from the centre of rotation to the blade tip. It is calculated as an integral with the r limits from 0 to 1. However, to obtain better quality results, a tip loss factor can be proposed [30]:(11)$$B=1-\frac{ac\left(r=1\right)}{2R}$$

The tip loss factor is taken as an upper limit of the coefficient integral. The thrust losses are also a result of the root cutout. It is usually 10% to 30% of the blade radius. Including tip loss factor and root cutout, the thrust integration is:(12)$$T=\frac{1}{2}\rho ab\Omega^{2}R^{2}\int_{r_{R}}^{B}c\left(\vartheta -\delta \right)r^{2}dr$$

According to [22], the mean lift coefficient for the main rotor blade can be determined:(13)$${\bar{C}}_{L}=\frac{6C_{T}}{\sigma }$$

The calculations described above were all prepared in MATLAB. The estimated mathematical thrust model is the basis for the CFD computation evaluation.

The program build is defined using the algorithm presented in Figure 3. The code is constructed as a loop to look for the collective angle required for a given aircraft weight.

It starts by defining the input parameters that can be obtained from first cut calculations and the mission flight conditions. In the beginning, the collective is positioned at 0° angle. Next, the position radius coordinates are defined for the twist and chord values in the given sections. The thrust is a radius position integral, and therefore, all program functions are defined as dependent on x. Using the chord and twist values, the chord and twist polynomials are interpolated. These polynomials are also used in the GRIP program to create the blade geometrics. Next, the inflow angle, the rotor solidity, and tip loss factor are given. All the specified equations are inputs to the thrust integral which is bounded by the root cutout and the tip loss factor. The blade attack angle is calculated using the pitch angle with addition of collective. The thrust equation is computed using loop. The collective angle is increased in each step until the result is similar to the hover required force for the assumed mass. The determined collective is used for the CFD analysis to check the parametric model aerodynamic parameters. The procedure described above can be applied for the hover case calculations and for vertical or horizontal flight conditions as well.

To present the possibilities of Matlab calculations for future main rotor optimization, a multi-dimensional analysis was conducted. The main rotor thrusts were calculated for different numbers of blades, twist angles, and mean chord values. The computations were made for the collective angle 7.47°. The distribution of thrust values as a function of twist angle and chord value are shown in Figure 4, Figure 5 and Figure 6. As predicted, the highest thrust value is obtained for a five-blade rotor, although the reference system is a four-blade rotor, corresponding to the rotor design of the Polish W-3 Sokol helicopter.

### 2.2. Modeling the Blade Geometrics and Aerodynamics Using Proprietary Software Applications

The calculations of the blade were performed for an existing main rotor blade of a Polish military W-3 SOKOL helicopter. The model could be easily modified with different parameters. The technical specifications are shown in Table 1, however, some details such as the blade and tip shape (which are provided from design documentation) cannot be published. The simulation was performed for the aerodynamic calculations, therefore, it was modeled as an empty shell only to provide the blade shape.

Figure 7 shows the original algorithm dedicated to the rotor blade parametric modeling. The rotor blade parametric models were prepared in Siemens NX Open Grip language. It is a programming language that can draw a CAD model with commands and is able to execute advanced customized operations in a more effective way than interactive NX operations. Numerous interactive operations can be executed using GRIP language. It is possible to draw geometric objects using control of parameters or load data from a file to modify the geometrics. Exemplary program code for parametric modeling of the blade geometrics is presented in Listing 1.
**Listing 1.** Part of GRIP code with blade surface generation.….SPL3(K+1)=SPLINE/PT0(1..N-1) $$ upper surfaceSPL4(K+1)=SPLINE/PT1(1..N-1) $$ lower surfacedfi=fi(4)*cr+fi(3)*cr+fi(2)*cr+fi(1)MAT=MATRIX/XYROT,-dfiSPL1(K+1)=TRANSF/MAT,SPL3(K+1)SPL2(K+1)=TRANSF/MAT,SPL4(K+1)$$ LN(K+1)=TRANSF/MAT,LN1(K+1)PT2(K+1)=POINT/ENDOF,XSMALL,SPL1(K+1)PT3(K+1)=POINT/ENDOF,XLARGE,SPL1(K+1)PT4(K+1)=POINT/ENDOF,XLARGE,SPL2(K+1)LN(K+1)=SPLINE/PT3(K+1),PT4(K+1)DELETE/PT0,PT1,SPL3,SPL4,LN1IFTHEN/K<MK=K+1JUMP/L10:ELSEJUMP/L40:ENDIFL40:A=&POINT(PT3(1))B=&POINT(PT4(1))SPLC(1)=SPLINE/PT2(1..K+1)SPLC(2)=SPLINE/PT3(1..K+1)SPLC(3)=SPLINE/PT4(1..K+1)SSRF(1)=BSURF/MESH,SPLC(1..2),WITH,SPL1(1..K+1),TYPE,3,TOLER,.01,.01SSRF(2)=BSURF/MESH,SPLC(1),SPLC(3),WITH,SPL2(1..K+1),TYPE,3,TOLER,.01,.01SSRF(3) = RLDSRF/SPLC(2),,SPLC(3)SSRF(4) =RLDSRF/SPL1(1),,SPL2(1)SSRF(5) =RLDSRF/SPL1(K+1),,SPL2(K+1)BLD(1)=SEW/SSRF(1..5)….

The program prepared for generating the main rotor blade was easy to modify, and the blade parameters could be changed without needing to repeat an interactive method. The blade radius, chord, chord distribution, twist, and airfoil shape could be easily changed to generate a model with new features. The process of parametric model generation is presented in Figure 8 and Figure 9.

### 2.3. Aerodynamic Modeling of a Single-Blade and Complete Main Rotor

The generated model was implemented in two computer environments to evaluate the properties of the obtained blade shape. The main rotor blade was implemented in a CFD software, i.e., ANSYS Fluent and an aerodynamic modeling software, i.e., DARcorporation Flightstream. The domain for CFD was prepared in ANSYS Spaceclaim. The air density in the simulations was 1.25 kg/m^3^ and it was constant.

The implementation was conducted to assess the model and to prepare its applicability in further research; however, it was also conducted to compare two different tools for aerodynamic rotorcraft analysis with mathematical calculation. With the results of the CFD analysis, a further strength analysis is possible. In addition, the CFD analysis is a crucial step in the optimization process of a modern main rotor blade design.

#### 2.3.1. One Blade CFD Modeling in ANSYS Fluent

In ANSYS Fluent, a one blade CFD model was prepared. The fluid domain was prepared using Ansys Spaceclaim. It was generated with a quarter of a sphere and half of a cylinder. The air enclosure were 20 m long and high, and the width of the enclosure was 10 m. A hybrid mesh was built for the computational area using the ANSYS Mesh module. Body and face sizing were used with inflation to generate the correct mesh. The boundary layer was modeled using the full thickness option. It was set at 20 mm for 25 layers with a growth ratio of 1.2. The obtained y+ value was 8. The maximum element size was set at 1000 mm, with a size reduction closer to the blade of 20 mm. The 3D bodies were transformed into a tetrahydra mesh with prismatic components within the boundary layer. The mesh consisted of almost 2,837,051 elements. The mesh is shown in Figure 10.

The boundary conditions are presented in Figure 11. The blade is fixed in the enclosure which is modeled by combining half of a hemisphere with a cylinder. In addition, to obtain correct body sizing, a smaller cuboid enclosure was made around the blade, and by naming the object faces, an air flow direction was projected.

To imitate the main rotor blade working environment, velocity change, pitch, and inflow angle were implemented with mathematical expressions. A velocity change was realized by an expression that raised the inflow speed with the span. To model a change in the inflow angle, the inflow directions on the *X* and *Y* axes were expressed using the cosine of the *X* axis and sine of the *Y* axis. The inflow angle was calculated from the inflow ratio with the Matlab program. Table 2 presents the inflow input function to model the air flow over the blade. The polynomial was a 4th grade polynomial, the higher grades gave similar results. The functions values changed with the span.

The viscosity model which was used for simulations was k-omega SST. It was applied because it is recommended for rotating machinery.

#### 2.3.2. One Blade Panel Modeling in FligtStream

In the second aerodynamic environment, a main rotor blade model was also prepared. The parametric blade model was easily implemented. Software was used to import the CAD files from commercial software, and therefore, the tool could be used in the optimization loop.

In the Flightstream, the mesh is generated only on the studied element. It does not required from user preparing an enclosure and environment. The mesh is generated automatically by the software. Because of the simplicity of the software, the inflow angle and the change of the pitch angle were modeled with the blade position in the reference frame. Velocity change was simulated by the build “shear” freestream option and prepared input .txt file with the velocity magnitude across the span. The advantage of the solver is the fact that the model for conducting a simulation is ready to use three–four times faster than in the Fluent environment. However, rotorcrafts are complex constructions that work in versatile conditions, and therefore, some working states may not be imitated. The solver was setup with steady viscous parameters. The solver setup is shown in Figure 12.

#### 2.3.3. Main Rotor CFD Modeling in ANSYS Fluent

In the second stage of the research, the blades were multiplied and positioned in a rotational plane. The fluid domain were also modeled in Spaceclaim. A mesh was built for the computational area using the Mesh Ansys module. The boundary conditions are presented in Figure 12. Body sizing and inflation were used to generate the correct mesh. The boundary layer was modeled using the full thickness option. The air enclosure was 75 m long and wide, and the height of the enclosure was 25 m. The maximum element size was set at 2000 mm, with a size reduction closer to the blade of 20 mm. The boundary layer was modeled using the full thickness option. It was set at 20 mm for 25 layers with a growth ratio of 1.2. The obtained y+ value was 8. The 3D bodies were transformed into a triangular mesh. The mesh consisted of nearly 5,300,000 elements.

The blades were placed in the enclosure which was modeled with cuboid. In addition, to obtain correct body sizing, a smaller cylinder enclosure was made around the blades. By naming the object faces, an air flow direction was projected. The analysis was performed with mesh motion, and therefore, there was no inlet, because the air flow was provoke by the rotating blades. The boundary conditions are presented on Figure 13.

The blades were set at the pitch angle calculated for the required thrust. The models were set to rotate with the main rotor angular speed. The inflow angle was assumed to be generated in the solver with the rotational movement. The viscosity model for the simulation that produced the most accurate results with the lowest time consumed was realizable k-epsilon with scalable wall function.

#### 2.3.4. Main Rotor Panel Modeling in FligtStream

Since the modeling was performed using CFD software, the panel environment was also tested to provide results for parametric modeling in the optimization procedure. The model building was more time-consuming and it was easy to make a mistake, because the pitch angle and inflow angle were modeled by the blade’s position. The blades were multiplied and positioned in a rotational plane. The mesh was generated the same way as it was generated for the one blade case. The cases were calculated with an unsteady solver. The rotational movement was modeled with motion settings, where the angular velocity and rotation axis could be defined.

## 3. Results

The evaluation results are shown in Table 1 and Table 2. In both cases, the parametric model has no issues when implemented. Calculations of the aerodynamic thrust were conducted for selected design weight values corresponding to those from the range of operating weights of the W-3 Sokol helicopter (i.e., between 4900 and 6400 kg).

Table 3 shows the comparison of the results of Fluent and Flightstream stationary blade modeling with mathematical calculations for a given thrust and the required collective angle for that thrust. The collective angles were calculated using thrust Equation (9) using Matlab procedure, combining the blade element theory with the momentum theory. According to the obtained results, shown in Figure 14, CFD modeling resulted in a high accuracy of expected resultant forces; both solvers produced similar results. However, for lower thrust, Flightstream, which is mainly designated to steady simulations, delivered a more accurate result.

During the preparation of the stationary simulation, it is crucial to model the exact inflow conditions. The main advantage of this type of imitation is the short time for the solver calculations. As compared with a four blade rotary model, it provides the results about 9–10 times faster. It also offers the possibility of checking the different collective values and inflow angle using one model and solver settings. This type of simulation is a good choice for evaluating the first cut blade loads.

Table 4 presents the comparison of results from Fluent and Flightstream for the main rotor model with modeled rotational movement. The mathematical assumptions are the same as in the steady model. The results presented in Figure 15 show that, for the rotary model, definitely better results are obtained with the Fluent solver. The Flightstream environment gives the most accurate findings with the unsteady solver, nevertheless, they is a 20% difference btween the mathematical and CFD solver model. The advantage of the Flighstream simulation is the duration of calculation. For the Ansys software, the calculations with mesh preparation last about 3 h (for a 30 threads computer station); with the DARcorporation solver, the simulation with meshing is ready in up to 30 min (with 8 threads). Despite the results, the Flightstream simulation can be used, when a designer needs to take a first look at the rotor aerodynamics behavior. However, to obtain accurate findings, a CFD simulation needs to be conducted.

The rotary model provides the calculations and forces that can be used in further analysis. It can be applied to the strength model and can calculate the required mass parameters to sustain the blade loads. Combining the models’ mass loads from rotary motion can also be simulated.

## 4. Conclusions

As stated in the Introduction, this work is part of an aerospace construction optimization research program. It is the second step in preparing the optimization loop for rotorcraft constructions.

In this research, it was confirmed that graphical programming language with a user-friendly syntax is a good solution for this type of geometric modeling. The commands are intuitive and easy to use. The code is short and the blade model is generated within a few seconds. In addition, the language offers an inertia analysis for solids, which is planned to be used in future research on the optimal blade design. The generated model was versatile and easy to adjust in different environments. There were no compatibility problems with the CFD solvers.

The parametric model behaved as it was assumed. The CFD analysis confirmed that the generated shape had the aerodynamic features that were preferred during the design phase. The mathematical assumptions of the inflow angle were confirmed with the rotary CFD model.

This research shows a new approach to rotor blade inflow modeling. The stationary blade with complex air flow analytical model gives correct results and consumes less computer power and time. Therefore, it is possibile to implement this type of simulation in the preliminary design of a main rotor, which is useful in optimization studies. Combining the stationary blade with parametric modeling is the first stage of optimization to evaluate a construction and to enhance the first cut shape. This method provides an initial shape in a short period of time. Therefore, even when the calculations are combined with mechanical analysis (FSI), the analysis time is reduced. Finally, the iterative procedure for optimization studies is proposed in Figure 16. In the first phase, the CFD model can be replaced with a panel code environment. To summarize, the results of this study can serve as a basis for developing a main rotor aerostructural model (combining aerodynamic properties and finite element structure) for further strength analysis, where both the aerodynamic and weight loads can be considered. In future work, in accordance with the research program, the CFD and FE model will be combined using scripting methods to prepare an optimization loop to find a design shape that fulfils modern combat field requirements and offers the best mass properties.

## Figures and Tables

**Figure 1 materials-15-04275-f001:**
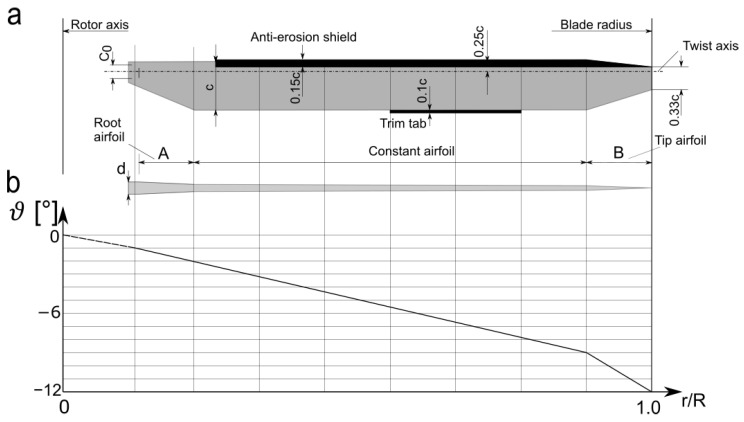
The geometry of the main rotor blade: (**a**) Rotor shape (*c*, chord and *c*_0_, mounting holes spacing); (**b**) blade parameters (*α_S_*(*r*/*R*), geometric twist angle in dependence on nondimensional radius; A and B, airfoil change zone; *d*, blade thickness) (figure prepared by authors).

**Figure 2 materials-15-04275-f002:**
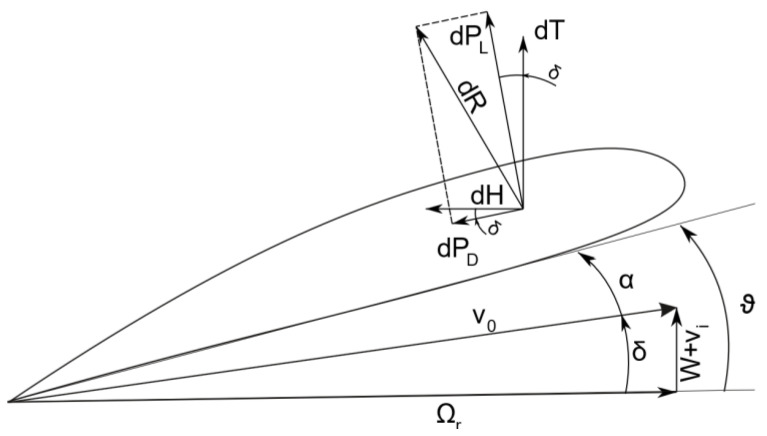
Blade element aerodynamic forces (figure prepared by the authors).

**Figure 3 materials-15-04275-f003:**
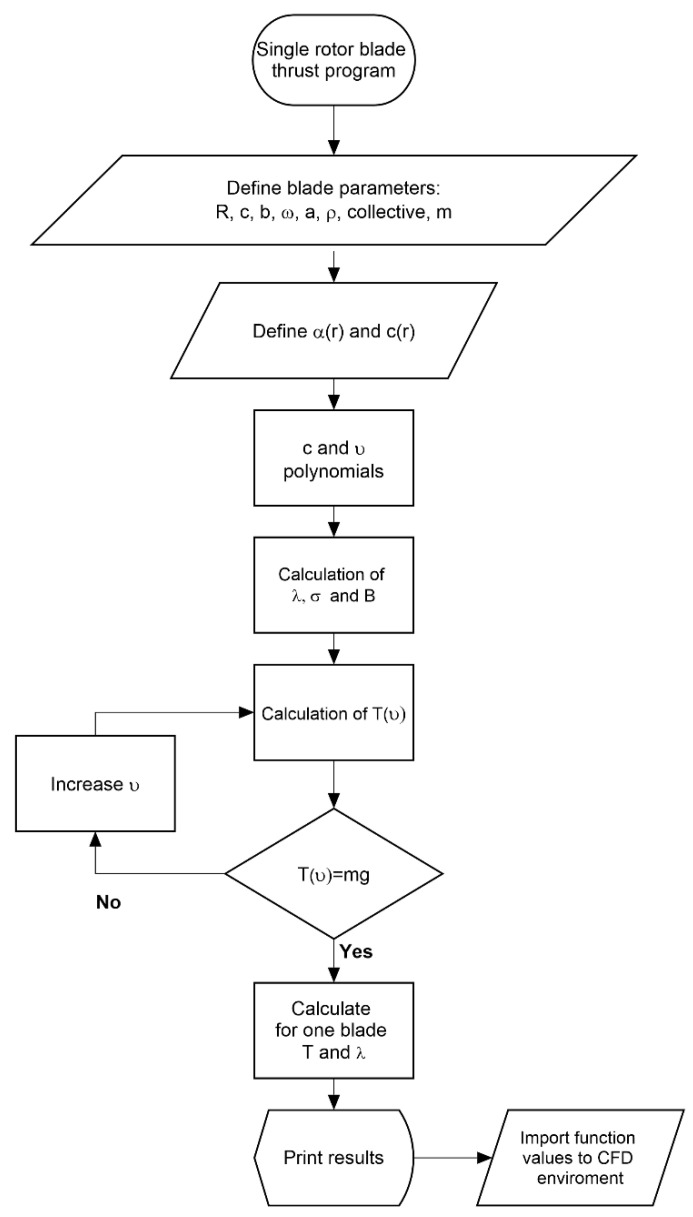
The algorithm developed in Matlab for determining the distribution of the blade’s parameters and its aerodynamic thrust.

**Figure 4 materials-15-04275-f004:**
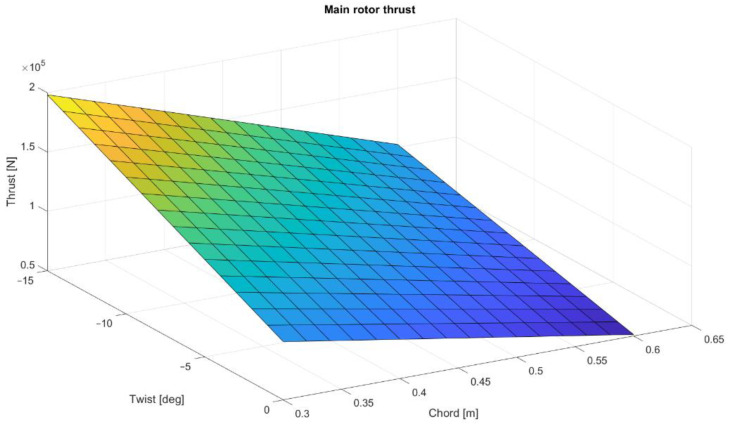
Three-blade rotor thrust distribution as a function of twist and mean chord.

**Figure 5 materials-15-04275-f005:**
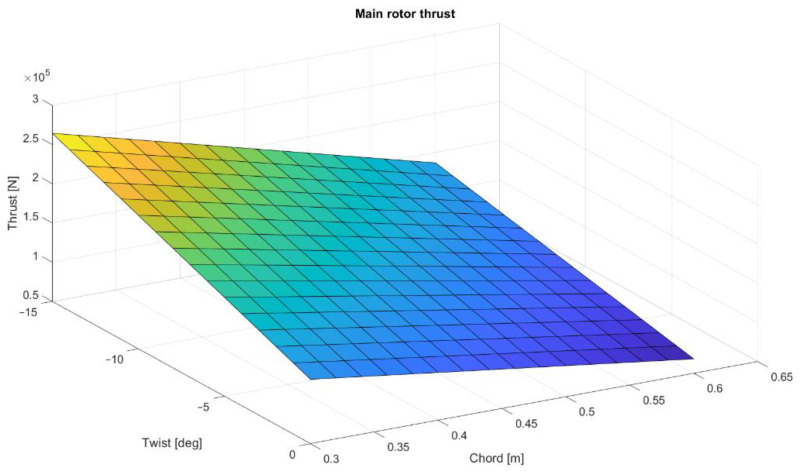
Four-blade rotor thrust distribution as a function of twist and mean chord.

**Figure 6 materials-15-04275-f006:**
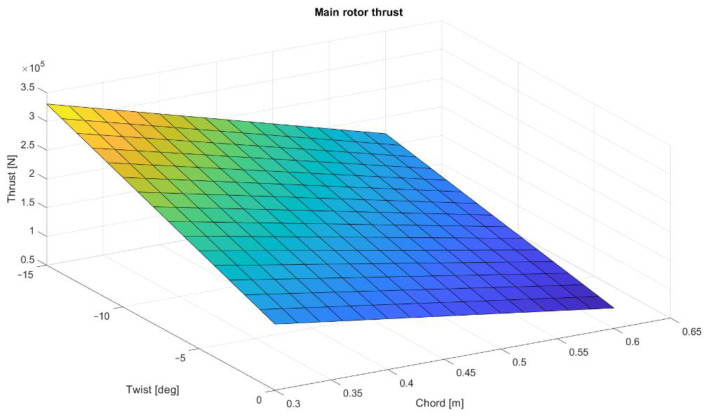
Five-blade rotor thrust distribution as a function of twist and mean chord.

**Figure 7 materials-15-04275-f007:**
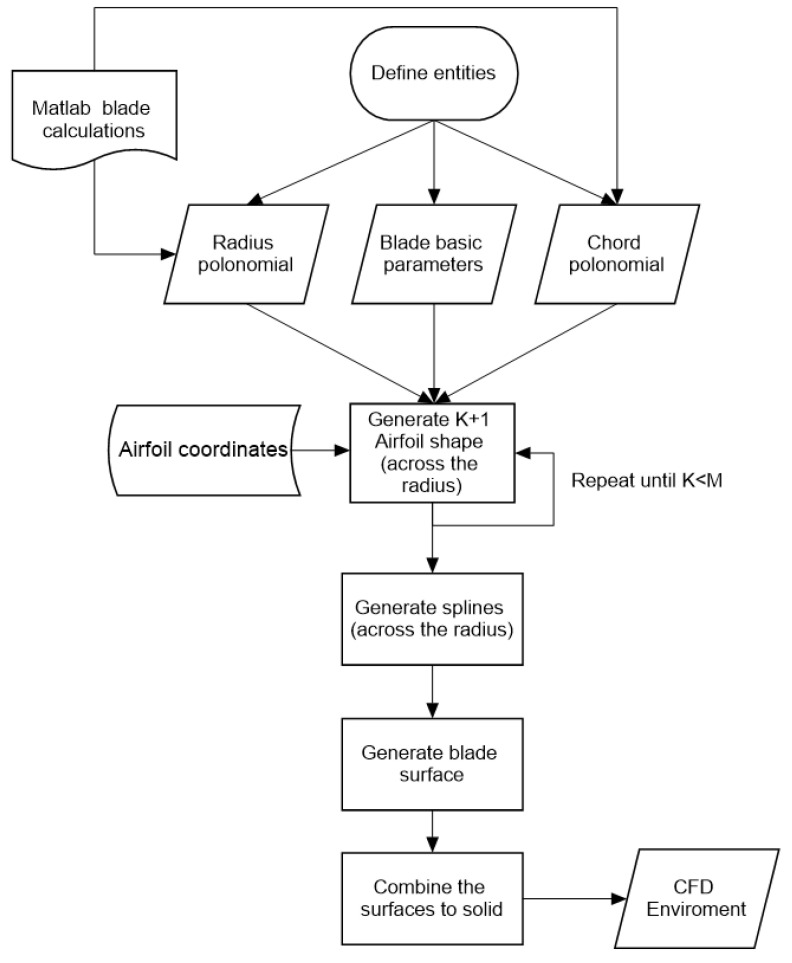
Diagram of the algorithm for parametric programming of the main rotor blade geometrics.

**Figure 8 materials-15-04275-f008:**
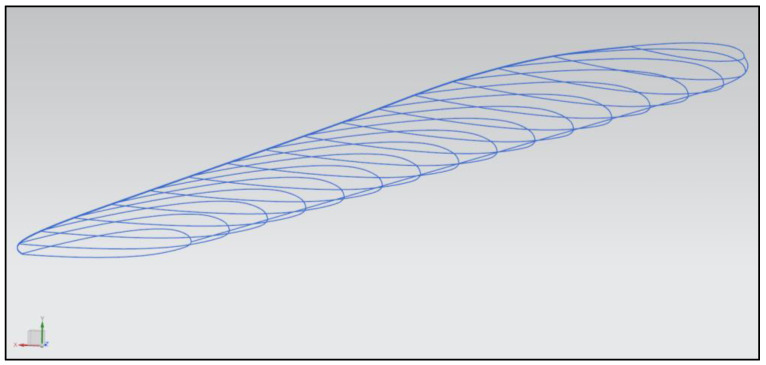
Main rotor blade shape curves.

**Figure 9 materials-15-04275-f009:**
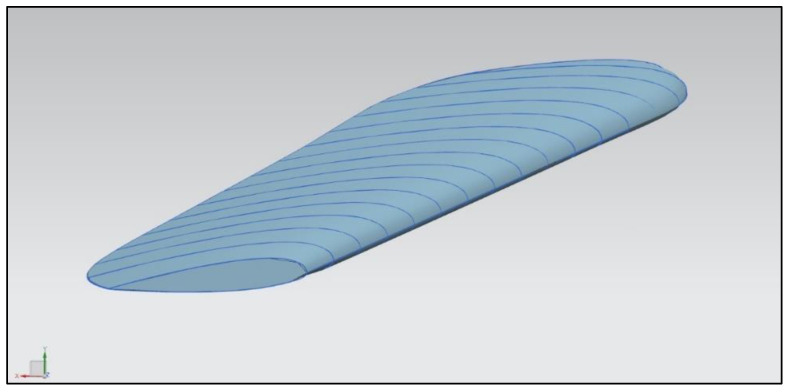
Model generated with defined curves.

**Figure 10 materials-15-04275-f010:**
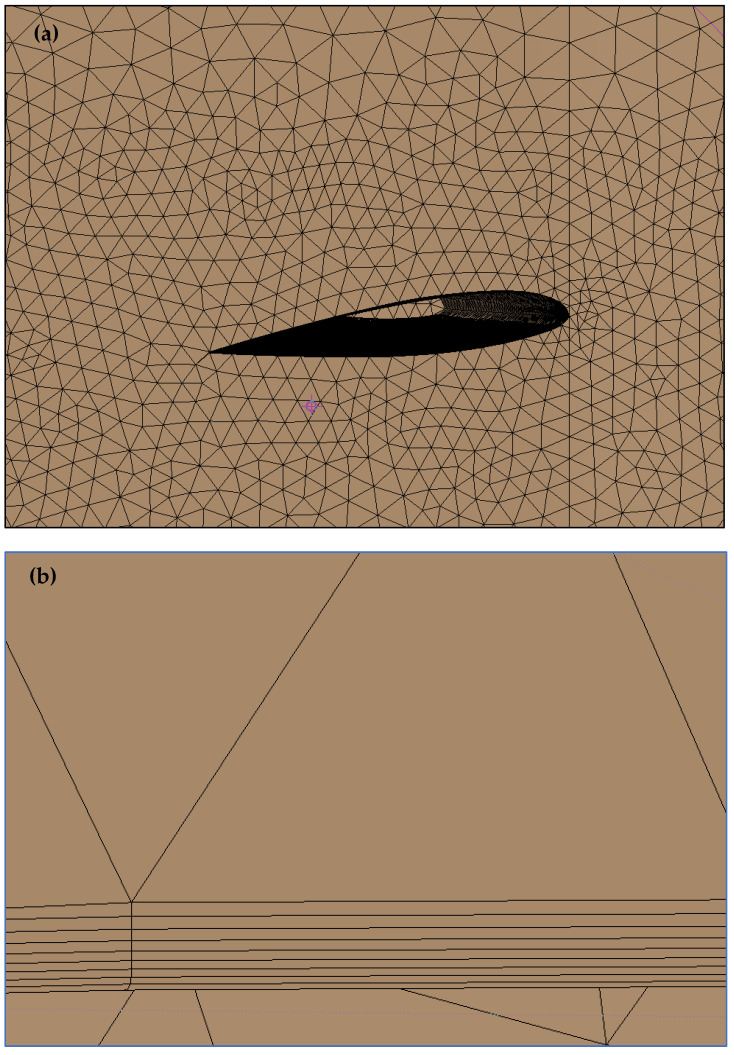
Rotor blade mesh with boundary layer: (**a**) Mesh; (**b**) boundary layer.

**Figure 11 materials-15-04275-f011:**
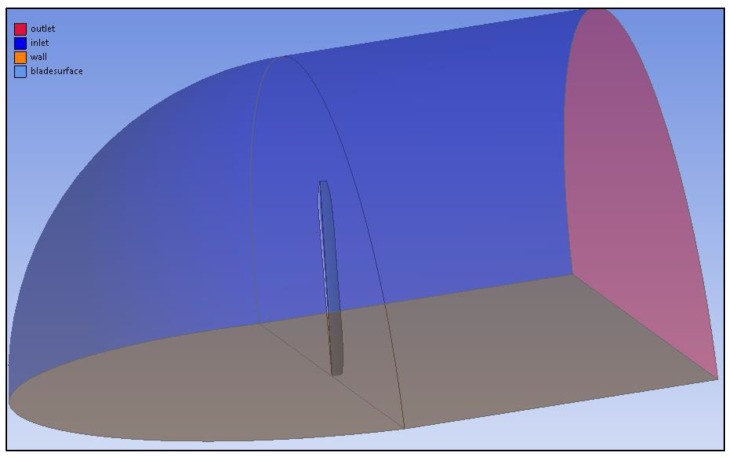
Single-blade model boundary conditions.

**Figure 12 materials-15-04275-f012:**
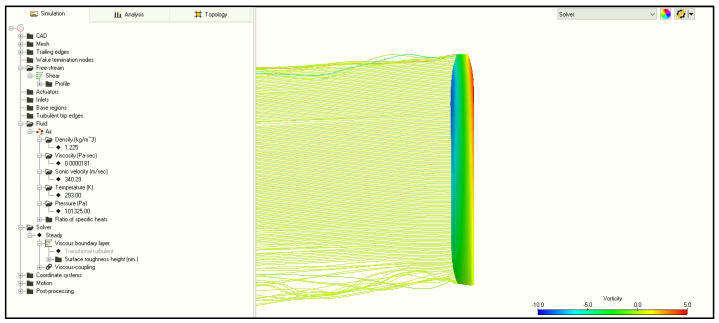
One blade panel code model with solver setup.

**Figure 13 materials-15-04275-f013:**
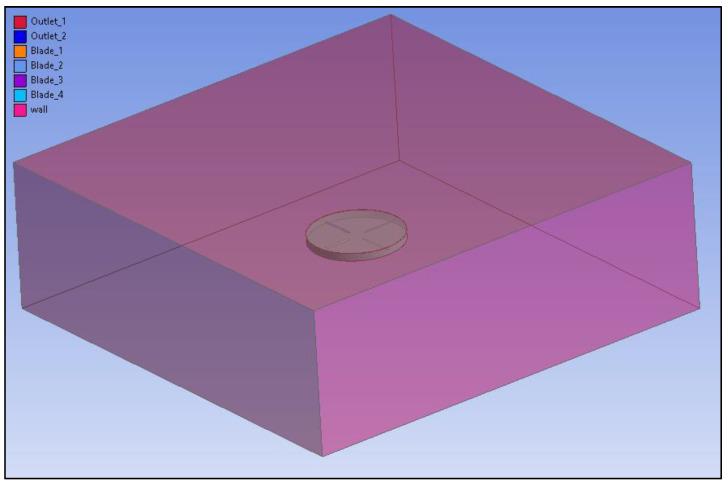
Four blade model boundary conditions.

**Figure 14 materials-15-04275-f014:**
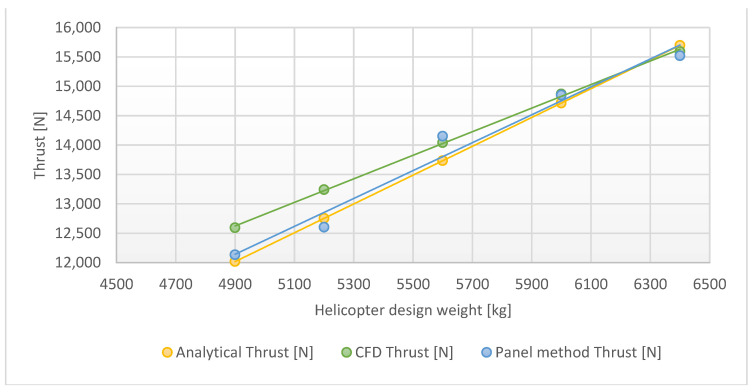
Line graphs of aerodynamic thrust dependencies for the one blade model.

**Figure 15 materials-15-04275-f015:**
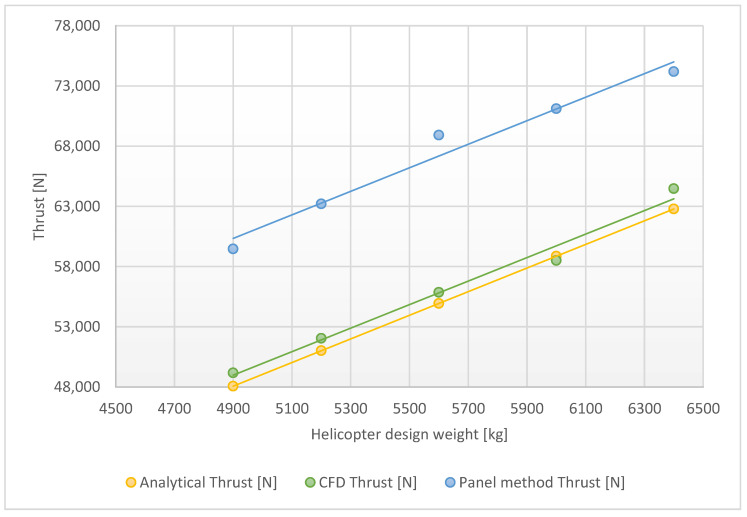
Line graphs of aerodynamic thrust dependencies for the complete rotor model.

**Figure 16 materials-15-04275-f016:**
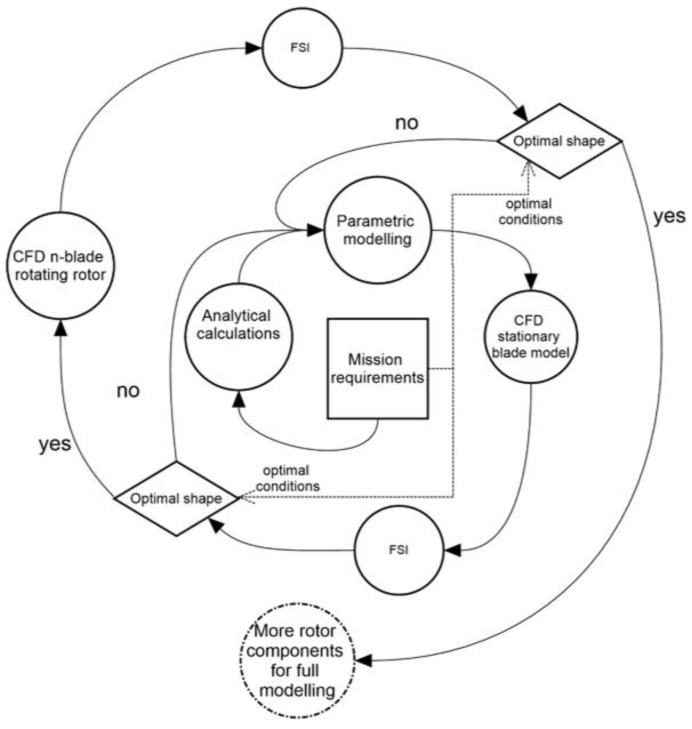
Graphical proposal of design spiral with results of the studies.

**Table 1 materials-15-04275-t001:** Main rotor technical specifications.

Rotor radius	7.85 m
Blade mean chord	0.44 m
Blade airfoil	NACA 23015
Geometric twist	−12°
Root cutout	0.21R
Root incidence angle	5.34°

**Table 2 materials-15-04275-t002:** Example of the inflow functions for the inflow modeling in the FLUENT rotor blade analysis.

Inflow *X* direction function for 5.96 pitch angle	cos(PI/180 × (5.96 − (−1.40420898843306 × 10^−5^ × (z/7.85[m])^3^ + 0.00154624103083247 × (z/7.85[m])^2^ − 0.0591574921598430 × (z/7.85[m]) + 1.07777970150355)))
Inflow *Y* direction function for 5.96 pitch angle	sin(PI/180 × (5.96 − (−1.40420898843306 × 10^−5^ × (z/7.85[m])^3^ + 0.00154624103083247 × (z/7.85[m])^2^ − 0.0591574921598430 × (z/7.85[m]) + 1.07777970150355)))

**Table 3 materials-15-04275-t003:** Results obtained from analyses of the one blade model.

Simulation Input Parameter			Fluent		Flightstream	
Weight (kg)	Analytical Thrust (N)	Collective (°)	CFD Thrust (N)	Δ (%)	Panel Method Thrust (N)	Δ (%)
6400	15,696	7.47	15,591	0.67%	15520	1.12%
6000	14,715	7.06	14,870	1.05%	14850	0.92%
5600	13,734	6.66	14,041	2.24%	14150	3.03%
5200	12,753	6.26	13,242	3.83%	12602	1.18%
4900	12,017	5.96	12,592	4.78%	12133	0.96%

**Table 4 materials-15-04275-t004:** Results obtained from analyses of the complete rotor model.

Simulation Input Parameter			Fluent		Flightstream	
Weight (kg)	Analytical Thrust (N)	Collective (°)	CFD Thrust (N)	Δ (%)	Panel Method Thrust (N)	Δ (%)
6400	62,784	7.47	64,472	2.69%	74186	18.16%
6000	58,860	7.06	58,504	0.60%	71112	20.82%
5600	54,936	6.66	55,862	1.69%	68917	25.45%
5200	51,012	6.25	52,044	2.02%	63205	23.90%
4900	48,069	5.95	49,175	2.30%	59462	23.70%

## Data Availability

Program code is available at: https://github.com/jakubkocjan/parametricrotordesign (accessed on 30 March 2022).

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
