# Peer review of "Helicopter Main Rotor Blade Parametric Design for a Preliminary Aerodynamic Analysis Supported by CFD or Panel Method"

_materials, 2022, doi:10.3390/ma15124275_

Round 1
Reviewer 1 Report
The authors propose a numerical design strategy for helicopter rotors. Many details are missing and the procedure cannot be reproduced with the information provided. As the authors state, the work is very preliminary and not complete. I suggest to finish the work to a complete procedure before publication.
Many details are missing that the method can be reproduced. Good aerodynamic predicitons with ANSYS require many settings, good mesh, ... . Nothing of that is described.
It would be preferable that scripts are shown as text rather than pictures (in figures 3 and 9). Further, variable declarations should end with a semiclon in Matlab.
I don't understand figure 17.
The mechanical aspect is neglected. The wake of one rotor can cause vibrations in the following rotor blade. Such phenomena are completely ignored.
There are several language errors, e.g. "It starts with defined the input parameters ..." -> "It starts with defining the input parameters ..." or "It starts with defined input parameters ...".
Author Response
Dear Sir/Madame,
We thank You for your review. I appreciate the time and effort that you have dedicated to provide your valuable feedback on my manuscript. I am are grateful for your insightful comments on my paper. I have been able to incorporate changes to reflect most of the suggestions provided by the reviewers. I have highlighted the changes within the manuscript. The corrected manuscript is attached to this submission.
The details have been update to ensure that the procedure can be reproduced. The source codes were published in the internet. Link is attached at the end of the manuscript
The parts of the scripts were presented as text. All off the code is provided via Github.
The figure 17 was deleted.
The more complex rotor analysis with mechanical analysis will be an object of the further research. Object of this research was to evaluate the parametric model with CFD and panel code environments for the main rotor optimisation procedures. In the next step the FSI analysis will be conducted.
The mistakes were corrected.
We look forward to hearing from you in due time regarding our submission and to respond to any further questions and comments you may have.
Your sincerely,
Jakub Kocjan

Reviewer 2 Report
These problems and comments are mentioned in the following:
- The English writing of the paper should be enhanced, and the paper should be reviewed to eliminate any present grammatical problems through the paper.
- In the Abstract, the methodology and purpose of the paper is described thoroughly; however, the results obtained from this research along with the validity of the suggested model is not explained. Therefore, it is of vital importance to mentioned the results achieved from the comparison of various models.
- In the part, where the numerical scheme is explained, it is of vital importance to mention the reasons of validity the chosen turbulence model for this case.
- In Introduction part,
- this part should include an extensive review of the studies performed previously in this field to represent and reveal the missed work and innovation of the current study. However, through the introduction of this paper, these considerations cannot be observed, and the innovation of the paper is not specified.
- Also, in the last sentence of the Introduction, it is written that, “This work is a part of a research program that is also aimed at finding the best aircraft construction optimization solutions.” According to this paper, it is helicopter construction or main rotor not aircraft!!
- About figures,
- For Figure 1 and 2, the used references should be mentioned in the caption.
- For figures 3 and 13, it is not appropriate for a scientific paper that the figure of the code written in the software environment be illustrated.
- The details of the CFD numerical simulation and panel method should be mentioned. This is necessary to make sure that if the results of the paper ae valid, and if the developed method is appropriate
- The paper should have a nomenclature.
Author Response
Dear Sir/Madame
We thank You for your review. I appreciate the time and effort that you have dedicated to provide your valuable feedback on my manuscript. I am are grateful for your insightful comments on my paper. I have been able to incorporate changes to reflect most of the suggestions provided by the reviewers. I have highlighted the changes within the manuscript. The corrected manuscript is attached to this submission.
Here is a point-by-point response to the comments and concerns.
The english were tried to be enhanced.
The abstract is updated with the results and the proposal of application.
The choose of turbulence model were explained.
The introduction were enhanced with the description of the studies performed previously and the innovation of the research.
The aircraft mistake were corrected.
The referencre for Figure 1 and 2 were added.
The source codes were published in the internet. Link is attached at the end of the manuscript The parts of the scripts were presented as text. All off the code is provided via Github.
The details of CFD simulation and panel method were described.
The symbols used in mathematical expression are explained under each expression
We look forward to hearing from you in due time regarding our submission and to respond to any further questions and comments you may have.
Your sincerely,
Jakub Kocjan

Reviewer 3 Report
This work is to provide some methods and design solutions for helicopter main rotor multidisciplinary optimization. But the manuscript in current status is more like a report, lack of scientific soundness. e.g. the Matlab program is important for calculation and generation, but the theoretical formula is more necessary for the method description in manuscript, with the program could be used as appendix. The same problem with the numerical setting (FIg. 13), the screenshot of the commercial software is not recommended. Moreover, some figures have low quality, such as Fig. 10, 11, 12, 14 and so on. As a research work, it should not only show the results captured from the software, but also present some understanding of the mechanism. In this case, this paper is not recommended to be published in current status.
Author Response
Dear Sir/Madame
We thank You for your review. I appreciate the time and effort that you have dedicated to provide your valuable feedback on my manuscript. I am are grateful for your insightful comments on my paper. I have been able to incorporate changes to reflect most of the suggestions provided by the reviewers. I have highlighted the changes within the manuscript. The corrected manuscript is attached to this submission.
Here is a point-by-point response to the comments and concerns.
The source codes were published in the internet. Link is attached at the end of the manuscript The parts of the scripts were presented as text. All off the code is provided via Github.
The mathematical formula were described in the paper.
The introduction were enhanced with the description of the studies performed previously and the innovation of the research. The conclusion were also developed.
The details of CFD simulation and panel method were described. The choose of turbulence model were explained.
Some Figures that were redundant were deleted.
We look forward to hearing from you in due time regarding our submission and to respond to any further questions and comments you may have.
Your sincerely,
Jakub Kocjan

Round 2
Reviewer 1 Report
The authors intend to present a new design approach for helicopter rotor blades.
The novelty of the manuscript is not existant or too low. The procedure described is lower than the standard in industry. It is described how to obtain a simple rotor blade, without consideretion of particular blade tips, thrust distribution, or strength considerations.
There are several language errors. Please fix them. For example, "The result is a compare of ..." should say "The result is a comparison of ..."
Author Response
Dear Sir/Madame
I thank You for additional your review. I appreciate the time and effort that you have dedicated to provide your valuable feedback on my manuscript. I am grateful for your insightful comments on my paper. I have been able to incorporate changes to reflect most of the suggestions provided by the reviewers. I have highlighted the changes within the manuscript. The corrected manuscript is attached to this submission.
The procedure description were updated. The blade and mesh details were provided.
The works shows the possibilities of the combined methods. The strength analysis, with FSI (fluid structure interaction) methods will be an object of further work.
The mistakes were corrected.
I look forward to hearing from you in due time regarding our submission and to respond to any further questions and comments you may have.
Your sincerely,
Jakub Kocjan
Reviewer 2 Report
​I appreciate the writers effort for eliminating the problems of the written paper in its revision. All the mentioned comments are addressed well and the paper is revised in a way that it is suitable to be published in this journal.
Author Response
Dear Sir/Madame
Once again I thank You for your review. I appreciate the time and effort that you have dedicated to provide your valuable feedback on my manuscript. I am grateful for your insightful comments on my paper.
Your sincerely,
Jakub Kocjan
Reviewer 3 Report
The author has conducted much work to improve the manuscript, but it's still lack of scientific soundness, with some unprofessional expressions, e.g. list of matlab code is not necessary in the content, figures of flow chart (fig. 3, 7, 15, etc) can be improved, figures of simulation results should be improved (fig. 11 etc.) instead of capture of the software. In this case, the highlights of this paper cannot be well presented.
Author Response
Dear Sir/Madame
I thank You for your additional review. I appreciate the time and effort that you have dedicated to provide your valuable feedback on my manuscript. I am are grateful for your insightful comments on my paper. I have been able to incorporate changes to reflect most of the suggestions provided by the reviewers. I have highlighted the changes within the manuscript.
The text and the description were improved to highlight the main aims and results of the research.
The matlab code were deleted.
The flow charts were improved.
The figure 11 shows the solver setup, however an mesh and simulation parameters were additionaly provided.
I look forward to hearing from you in due time regarding our submission and to respond to any further questions and comments you may have.
Your sincerely,
Jakub Kocjan